# The Acute Effects of Normobaric Hypoxia on Strength, Muscular Endurance and Cognitive Function: Influence of Dose and Sex

**DOI:** 10.3390/biology11020309

**Published:** 2022-02-15

**Authors:** Raci Karayigit, Mustafa Can Eser, Fatma Nese Sahin, Cengizhan Sari, Angela Sanchez-Gomez, Raul Dominguez, Mitat Koz

**Affiliations:** 1Faculty of Sport Sciences, Ankara University, Gölbaşı, Ankara 06830, Turkey; mceser@ankara.edu.tr (M.C.E.); nesesahin@ankara.edu.tr (F.N.S.); 2Faculty of Sport Sciences, Muş Alparslan University, Muş 49001, Turkey; c.sari@alparslan.edu.tr; 3Department of Nursing Pharmacology and Physiotherapy, Faculty of Medicine and Nursing, University of Córdoba, 14000 Córdoba, Spain; asgomez@uco.es; 4Departamento de Motricidad Humana y Rendimiento Deportivo, Universidad de Sevilla, 41013 Sevilla, Spain; rdherrera@us.es; 5Physiotherapy and Rehabilitation Department, Faculty of Health Sciences, Eastern Mediterranean University, North Cyprus, Mersin, Famagusta 99628, Turkey; mitat.koz@emu.edu.tr

**Keywords:** hypoxic dose, resistance training, muscular endurance, sex difference, flanker

## Abstract

**Simple Summary:**

Hypoxic training is a novel method to increase resistance training adaptations. There is some evidence that resistance exercise performed in systemic hypoxia can lead to structural and functional adaptations of skeletal muscle. Studies have also demonstrated that normobaric hypoxia (i.e., normal pressure, low oxygen) increases intramuscular metabolic stress and type two (fast twitch) fiber recruitment, which leads to a greater morphological adaptations over time. However, to date, there is no research that has investigated the effects of different doses of acute normobaric hypoxia on strength and muscular endurance performance, nor has there been research investigating potential sex-based differences.

**Abstract:**

The aim of this study was to examine the acute effects of different levels of hypoxia on maximal strength, muscular endurance, and cognitive function in males and females. In total, 13 males (mean ± SD: age, 23.6 ± 2.8 years; height, 176.6 ± 3.9 cm; body mass, 76.6 ± 2.1 kg) and 13 females (mean ± SD: age, 22.8 ± 1.4 years; height, 166.4 ± 1.9 cm; body mass, 61.6 ± 3.4 kg) volunteered for a randomized, double-blind, crossover study. Participants completed a one repetition strength and muscular endurance test (60% of one repetition maximum to failure) for squat and bench press following four conditions; (i) normoxia (900 m altitude; F_i_O_2_: 21%); (ii) low dose hypoxia (2000 m altitude; F_i_O_2_: 16%); (iii) moderate dose hypoxia (3000 m altitude; F_i_O_2_: 14%); and (iv) high dose hypoxia (4000 m altitude; F_i_O_2_: 12%). Heart rate, blood lactate, rating of perceived exertion, and cognitive function was also determined during each condition. The one repetition maximum squat (*p* = 0.33) and bench press (*p* = 0.68) did not differ between conditions or sexes. Furthermore, squat endurance did not differ between conditions (*p* = 0.34). There was a significant decrease in bench press endurance following moderate (*p* = 0.02; *p* = 0.04) and high (*p* = 0.01; *p* = 0.01) doses of hypoxia in both males and females compared to normoxia and low dose hypoxia, respectively. Cognitive function, ratings of perceived exertion, and lactate were also significantly different in high and moderate dose hypoxia conditions compared to normoxia (*p* < 0.05). Heart rate was not different between the conditions (*p* = 0.30). In conclusion, high and moderate doses of acute normobaric hypoxia decrease upper body muscular endurance and cognitive performance regardless of sex; however, lower body muscular endurance and maximal strength are not altered.

## 1. Introduction

Coaches and athletes are always looking for innovative training methods to gain an advantage. Since the 1968 Mexico Olympic games, hypoxic training has become popular especially among endurance athletes using Live High/Train Low (LHTL) or Live High/Train High (LHTH) strategies [1]. Beyond traditional LHTL or LHTH paradigms, a novel method of simulated altitude training entitled Live Low/Train High (LLTH) is becoming more popular for athletes [2]. Such an approach has been used to increase sea level exercise performance [3]. Recently, LLTH has been shown to augment physiological adaptations following resistance exercise, possibly via alterations in metabolic stress and greater intramuscular responses [4,5]. However, it is important to note that there is limited research examining responses to hypoxia following resistance exercise compared to aerobic or high intensity interval training.

Acute exposure to hypoxia may negatively alter resistance exercise performance. Mechanistically, these negative effects may be associated with diminished muscular and arterial oxygenation, electromyographic activity [6], neuromuscular activation [7], peak velocity [8], and increased expiratory parameters [8]. However, from a physiological perspective, the acute metabolic and neuromuscular stressors may augment hypertrophy over time [4,9,10,11,12]. Gains in muscle mass appear to be due to an enhancement in the recruitment of type II fibers, since type I fibers fatigue more quickly in a hypoxic environment [11]. Further, hypoxia-inducible factor-1 (HIF-1), a transcription factor, is upregulated, and is important to promote a slow to fast fiber-type transition in skeletal muscle [13]. However, the most probable mechanism by which hypoxia augments resistance training responses is due to elevated metabolic stress [4,9], leading to increases in motor unit recruitment [4]. The accumulation of metabolites [14,15] increases plasma growth hormone [16] and muscle cell swelling [10], which may be associated with elevated muscle protein synthesis.

In theory, the altered physiological and metabolic responses and acute performance changes may vary according to the dose of hypoxia. Presently, there is a lack of research investigating the optimal hypoxic dose required during strength training [17]. Campo et al. [18,19] reported that peak-mean power, RPE, and blood lactate during a resistance circuit training was significantly different in high hypoxia (F_i_O_2_ = 0.13; ~3.800) compared to moderate hypoxia (F_i_O_2_ = 0.16; ~2.100 m) and normoxia. In contrast, two studies [4,20] did not observe differences in force and muscular power during resistance exercise between high hypoxia (F_i_O_2_ = 0.13), moderate hypoxia (F_i_O_2_ = 0.16), and normoxia. Nonetheless, effort during resistance exercise could be greater in normobaric hypoxia than normoxia [21], which would influence performance. However, fixed number of repetitions was used in the test protocols of the aforementioned studies which hinder participants from reaching muscular failure. Further, resistance training to muscle failure could potentially increase hypertrophic adaptations by enhancing exercise-induced metabolic stress [22]. To date, only a few studies have investigated the effects of acute high and moderate hypoxia on leg extension [23], bench press [24], and biceps curl [25] exercise to failure. However, results were equivocal with studies showing either a negative impact of hypoxia [25] or no effect [23,24]. Further, these studies were only conducted on males [23,24,25] and untrained subjects [24].

Although relatively little is known about the acute responses to resistance exercise between sexes, females were reported to be more resistant to fatigue and faster to recover from fatiguing exercise than males in tasks utilizing low intensity loads and a slow repetition velocity in both concentric and eccentric phases [26]. Presently, there is no research examining sex-based responses to resistance exercise in hypoxia, which is warranted due to performance differences [27]. Females, compared to males, have greater fatigue resistance during training, and enhanced recovery, despite higher cardiovascular strain and RPE [28,29]. Females were also found to have lower lactate levels during hypoxic exercise (F_i_O_2_: 0.13), as well as higher glucose levels during recovery [30]. As such, the effect of hypoxia may differ in males and females.

Cerebral oxygen desaturation was significantly associated with impaired cognitive function (CF) [31]. As the severity of hypoxia increases, the level of deoxygenation increases. Moreover, two recent meta-analyses reported that attention, a parameter of complex cognitive processes, moderates acute muscular endurance [32] and strength [33] performance. However, currently there are no studies that have investigated CF before and after resistance exercise during hypoxia.

Therefore, the aim was to examine the acute effects of different doses of normobaric hypoxia (i.e., low [2000 m], moderate [3000 m], and high [4000 m]) on maximal strength, muscular endurance, and CF according to sex utilizing resistance exercise performed over multiple sets and to failure. We hypothesized that as the dose of hypoxia increased, maximal strength, muscular endurance, and CF will concomitantly decrease, while blood lactate and RPE will increase.

## 2. Materials and Methods

### 2.1. Participants

For this study, 13 males (mean ± SD: age, 23.6 ± 2.8 years; height, 176.6 ± 3.9 cm; body mass, 76.6 ± 2.1 kg) and 13 females (mean ± SD: age, 22.8 ± 1.4 years; height, 166.4 ± 1.9 cm; body mass, 61.6 ± 3.4 kg) who were healthy and were non-smokers volunteered to participate. The inclusion criteria included: (a) free from neuromuscular and musculoskeletal disorders, aged 18–30 years; and (b) able to perform a successful back squat and bench press exercise with load corresponding to 125% and 100% of their current body mass, respectively. Furthermore, participants had at least three years of resistance training experience, underwent training four times per week (which included squat and bench press in their routines), and were considered intermediately resistance trained [34]. All participants reported no previous exposure to an altitude of greater than 900 m within the last eight months, and were taking no substances that could affect the muscular and cognitive performance (i.e., creatine, anabolic steroids). Participants were requested to refrain from exercise, alcohol, and caffeine intake for 24 h before each session in order to maintain their customary sleep, to complete a 24 h dietary record before the first session, and to replicate this diet for 24 h before each subsequent session to standardize energy intake. Adherence to these instruction was verbally confirmed at the beginning of each session. This study was approved by Muş Alparslan University Scientific Research Ethics Committee (Approval no: 10776-6/31) and conducted in accordance with the Helsinki Declaration. Written informed consent was obtained from all participants prior to beginning the study.

### 2.2. Study Design

To investigate whether the degree of acute normobaric hypoxia affects strength (1RM), muscular endurance (repetitions to failure with 60% of 1RM), and cognitive function (Flanker Task; reaction time and response accuracy) in males and females, each participant performed the test protocol under four conditions with different O_2_ availability: (1) normoxia (NORM; 900 m altitude; F_i_O_2_: 21%); (2) low hypoxia (LowH; 2000 m altitude; F_i_O_2_: 16%); (3) moderate hypoxia (ModH; 3000 m altitude; F_i_O_2_: 14%); and (4) high hypoxia (HighH; 4000 m altitude; F_i_O_2_: 12%). The study used a randomized, double-blind crossover research design. During each experimental testing session, participants wore a face mask that was connected to a hypoxic generator (Everest Summit II, Hypoxico, New York, NY, USA), which controlled the oxygen availability. Level of altitude was observed on the hypoxic generator’s screen, which was hidden from participants to maintain blindness. To verify that hypoxic and normoxic conditions were provided, peripheral oxygen saturation (SpO_2_) was measured by a pulse oximeter (Hypoxico Oxycon, New York, NY, USA) attached to the participants’ finger. For this study, two familiarization sessions were conducted to ensure that the participants were able to squat and bench press with proper technique with the hypoxic generator mask. The testing sessions were well tolerated, and there were no reported side effects. In total, participants came to the laboratory on six occasions, which were separated by 72 h to allow for complete recovery. The first two sessions were familiarization sessions, which were similar to the experimental sessions, during which the participant’s performed the back squat on a smith machine (Esjim, Eskişehir, Turkey) and the bench press exercise on a rack with safety bar. All sessions were supervised by a certified personal trainer. During the familiarization and warm-up, the personal trainer provided technical feedback, then participants completed a 1RM testing protocol and three sets with 60% of 1RM to failure for both back squat and bench press, respectively, while wearing the hypoxic generator’s face mask. Bar grip position was recorded and replicated for subsequent sessions. These exercises were chosen to test lower and upper body’s major muscle groups and their common inclusion in resistance training programs. Participants were also familiarized with and practiced the cognitive function (CF) test until they achieved consistent scores. Additionally, participants were also introduced to the 6–20 Borg scale [35] for measuring their ratings of perceived exertion (RPE). All sessions took place in the afternoon (14.00–17.00) in order to minimize the diurnal influence on muscle strength. On arrival at the laboratory, resting measures of heart rate (HR) (Polar Team 2, telemetric system, Kempele, Finland), capillary lactate concentration (LAC) (lactate scout, USA), and CF measurements were taken. After 5 min passive rest while wearing hypoxic generator’s face mask for familiarization and 5 min warming-up on a treadmill, squat 1RM strength and 3 sets of 60% of 1RM muscular endurance tests were performed, respectively. Following 5 min passive rest, the same procedures were replicated for bench press. Immediately after the muscular test protocols, CF was measured again. HR, LA, RPE, and SpO_2_ were recorded at different time points during the testing protocol (Figure 1). Total hypoxia exposure duration was ~40 min.

#### RM Strength and 60% of 1RM Muscular Endurance Test Protocol

1RM strength performance was identified in three to five steps according to Baechle and Earle protocol [34]. After 10 repetitions with 20 kg weights was completed, participants then rested passively for 1 min, which was followed by a further three to five repetitions with an added 10% and 20% more weight for the bench press and squat exercises, respectively. Following a 2 min passive rest period, a weight near to their predicted 1RM was used, and participants performed two to three repetitions. Following another 3 min of passive rest, the weight was further increased by another 10–20% for the squat and 5–10% for the bench press if the participant successfully lifted the weight. If the lift was unsuccessful, the weight was decreased by 5–10% for the squat and 2.5–5.0% for the bench press for another 1RM attempt. Further, 3 min was given after 1RM establishment and the weight was lowered to 60% of 1RM; thereafter, three sets of muscular endurance tests with 60% of 1RM with a two minutes passive rest period was carried out by the participants. Movement tempo during muscular endurance test was standardized to two seconds for both concentric and eccentric phases via a metronome. Total repetition number to failure was used as a muscular endurance performance. The test protocol was terminated according to three criteria: (1) voluntarily completion the repetition; (2) unable to perform repetitions synchronously with the metronome for three consecutive repetitions; and (3) unable to lift with proper technique and posture.

### 2.3. Cognitive Function

A modified version of Flanker task was used to measure cognitive performance and run on a Dell Computer via appropriate software (InqusitLab 5.0, Milliseconds) [36]. A yellow fixation star appeared in the center of the screen, followed by five black arrowheads in a line which appeared for 200 milliseconds on a white screen background with a response window of 1000 milliseconds. For this, two congruent (< < < < <) and two incongruent (< < > < < or > > < > >) tasks were provided in equal probability and participants were instructed to react as fast and accurately as possible to the direction of the target arrow (in the middle) by pressing corresponding response buttons with their left or right index fingers. The interstimulus time gap varied from 1000 to 1500 ms, and the trial order was random for each participant. Following 20 practice trials, participants performed 100 trials while wearing earplugs. The total duration of the cognitive performance test was approximately three minutes. Mean response accuracy (%) and reaction time (ms) were recorded to measure cognitive performance.

### 2.4. Statistical Analysis

All analyses were conducted with IBM SPSS statistics for Windows, version 22.0 (IBM Corp., Armonk, NY, USA). Normal distribution was confirmed by the Shapiro–Wilk test. A three-way repeated measures analysis of variance (ANOVA) was performed to investigate the main effects for (1) condition, (2) sex, and (3) time or sets. Furthermore, three-way ANOVA examined the interaction effect of condition × sex × time or set. A Mauchly’s test was used to assess sphericity followed by the Greenhouse–Geisser adjustment, where appropriate. Relevant significant main or interaction effects were further examined in a Bonferroni post-hoc analysis. A 95% confidence interval (CI) was calculated, and partial eta squared (η^2^) were reported with significant ANOVA main effects as a measure of effect size as trivial (<0.10), moderate (0.25–0.39) or large (≥0.40) [37]. A two-way mixed model in consistency type of intraclass correlation coefficients (ICC) was calculated to determine the consistency of the four trials. Data is provided as mean ± SD and statistical significance was set at *p* < 0.05.

## 3. Results

### 3.1. Maximum Strength (1RM) and 60% of 1RM Muscular Endurance Performance

Hypoxia did not affect the 1RM squat (*p* = 0.33, η^2^ = 0.09) or bench press strength (*p* = 0.68, η^2^ = 0.04) (Figure 2). Although there was no main effect for condition in 60% of 1RM squat endurance (*p* = 0.34, η^2^ = 0.09), it was detected that there was a statistical trend in the third set between HighH and NORM in males (*p* = 0.059) and females (*p* = 0.085), and between ModH and NORM in females (*p* = 0.087). Furthermore, repeated measures with ANOVA detected significant main effect for bench press 60% of 1RM endurance performance (*p* = 0.01, η^2^ = 0.28). The Bonferroni post hoc test revealed that HighH (*p* = 0.01, 95% CI = −0.67–0.07; *p* = 0.01, 95% CI = −0.74–0.10) and ModH (*p* = 0.04, 95% CI = −0.75–0.02; *p* = 0.02, 95% CI = −0.81–0.06) had significantly lower repetition numbers during 60% of 1RM endurance test compared to LowH and NORM. Lastly, no significant condition × sex × set interaction was detected in the squat (*p* = 0.69, η^2^ = 0.05) and bench press (*p* = 0.96, η^2^ = 0.01) with 60% of 1RM endurance performance (Figure 3 and Figure 4). ICC values during three sets of 60% of 1RM squat endurance performance were 0.96 and 0.97 for males and females, respectively. In the bench press with 60% of 1RM endurance, ICC were 0.96 and 0.97 for males and females, respectively.

### 3.2. Cognitive Function

Response accuracy for the congruent condition was not significantly different between conditions (*p* = 0.56, η^2^ = 0.05). Further, no significant condition × sex × time interaction (*p* = 0.07, η^2^ = 0.18) was found. However, hypoxia affected reaction time for the congruent condition (*p* = 0.03, η^2^ = 0.21). Post-hoc analysis revealed that NORM was significantly faster than ModH (*p* = 0.04, 95% CI = −15.63–0.35) and HighH (*p* = 0.01, 95% CI = −0.67–0.07).

Response accuracy for the incongruent condition was not significantly different between conditions (*p* = 0.09, η^2^ = 0.17). However, hypoxia affected reaction time for the incongruent condition (*p* = 0.01, η^2^ = 0.43). The Bonferroni analysis demonstrated that NORM was significantly lower than ModH (*p* = 0.01, 95% CI =−22.77–3.86) and HighH (*p* = 0.01, 95% CI = −36.14–10.77). Furthermore, LH was also significantly different than HighH (*p* = 0.01, 95% CI = −29.26–4.85) (Table 1).

### 3.3. Heart Rate, RPE, Lactate and Oxygen Saturation

There was no difference between conditions (*p* = 0.30, η^2^ = 0.10) in heart rate. However, RPE values were significantly different between conditions (*p* = 0.01, η^2^ = 0.33). Post-hoc analysis showed that ModH (*p* = 0.02, 95% CI = 0.10–0.98) and HighH (*p* = 0.01, 95% CI = 0.31–1.51) were significantly higher than NORM. HighH was also significantly higher than LowH (*p* = 0.03, 95% CI = 0.05–1.44). There was a significant main effect on lactate between conditions (*p* = 0.01, η^2^ = 0.57). Post-hoc analysis showed that, after the bench press exercise, HighH was significantly higher than NORM (*p* = 0.01, 95% CI = 0.17–0.45) and LowH (*p* = 0.01, 95% CI = 0.17–0.38). ModH was also significantly higher than NORM (*p* = 0.01, 95% CI = 0.12–0.33) and LowH (*p* = 0.01, 95% CI = 0.11–0.27). Lastly, as expected, oxygen saturation values were significantly different between conditions (*p* = 0.01, η^2^ = 0.97): as hypoxic dose increases, in turn, oxygen saturation decreases (Table 2).

## 4. Discussion

To our knowledge, no study to date has examined the relationship between the effect of hypoxia dose and sex on acute resistance training performance. As such, the purpose of this study was to examine the acute effects of different doses of normobaric hypoxia on strength, muscular endurance, and cognitive function according to sex. The main findings demonstrate that moderating effects of hypoxia on strength, muscular endurance, and cognitive function did not differ between sex. Further, high (4000 m) and moderate (3000 m) doses of hypoxia significantly altered bench press endurance performance, cognitive function, RPE, and blood lactate levels. However, 1RM strength (both squat and bench press) and squat endurance performance was not influenced by hypoxia.

### 4.1. Strength (1RM) Performance

Our data indicate that 1RM squat and bench press strength performance was not different between conditions. This finding is consistent with Smith et al. [23], who concluded that moderate (F_i_O_2_ = 15.4%) and severe (F_i_O_2_ = 12.9%) hypoxia did not affect 1RM leg extension strength. Another study by Feriche et al. [38] reported that 1RM bench press strength did not show a difference between normobaric hypoxia (F_i_O_2_ = 15.7%) and normoxia. It seems that hypoxia has no influence on 1RM strength. Moreover, Girard et al. [25] alleged that, at higher training loads, the effects of hypoxia on muscular performance tends to be smaller. On the other hand, the method of strength measurement (1RM) in the current and previous [23,38] studies may be disputable due to not being sensitive enough to detect subtle effects of hypoxia on 1RM strength. It may be asserted that potential negative effects of hypoxia on 1RM strength can be lost in large inter-day 1RM strength variations [39]. In the present study, the ICC values during 1RM strength measurements showed high consistency (0.97–0.99) between conditions. Further studies are needed to examine the influence of hypoxia on strength performance measured via more sensitive devices, such as an isokinetic dynamometer or gauge.

### 4.2. Muscular Endurance (60% of 1RM) Performance

In the current study, high (F_i_O_2_ = 12%) and moderate (F_i_O_2_ = 14%) doses of hypoxia had a negative impact on 60% of 1RM bench press repetition to failure performance. However, low (F_i_O_2_ = 16%) dose hypoxia did not change neither squat nor bench press muscular endurance performance compared to normoxia. Hence, resistance exercise with high or moderate doses of hypoxia might develop more fatigue due to the increase in metabolic stress than the fatigue that develops in low hypoxia or normoxia (Table 2). This fact might be explained by exacerbated perturbations of cellular homeostasis in active skeletal muscles [19]. In this respect, high (F_i_O_2_ = 13%), but not low (F_i_O_2_ = 16%), doses of hypoxia increased lactate and diminished blood HCO_3_-, causing a reduction in blood pH [19]. This indicates that a high dose of hypoxia relies more on non-oxidative ATP phosphorylation [40] and increases intracellular acidosis, which are known to contribute to muscular fatigue [41].

Decrement in bench press performance with hypoxia in a dose-dependent manner was also shown by Campo et al. [18], who reported that peak and mean force were significantly lower in high dose hypoxia (F_i_O_2_ = 13%) when compared with low dose hypoxia (F_i_O_2_ = 16%) and normoxia. However, in the same study [18], a dose effect was not observed in the half squat exercise. Further, dose of hypoxia was previously shown as a moderating factor on muscular performance during high intensity resistance circuit training [19]. In support, a meta-analysis concluded that repetition to failure performance occurs earlier during resistance exercise with high but not low hypoxia [42]. Conversely, high (F_i_O_2_ = 13%) or moderate (F_i_O_2_ = 16%) hypoxic stimulus did not alter back squat and deadlift force or power variables [20]. Discrepancies in the type of test protocols (circuit training vs. one/two exercise; at an intensity between 60 vs. 85% 1RM) may vary the influence of various doses of hypoxia on resistance exercise performance.

The repetition to failure test protocol required participants to use maximum effort, thus, level of effort could be greater than protocols which use a fixed number of repetitions [4,18,19,20,38]. There are only a few studies that have used repetition to failure at a given percentage of 1RM to measure muscular endurance performance in hypoxia. For example, three sets of 70% 1RM leg extension to failure performance did not change between normoxia, moderate (F_i_O_2_ = 15.4%), and high (F_i_O_2_ = 12.9%) hypoxia [23]. Secondly, three sets of 75% 1RM to bench press failure performance was not different between normoxia and high hypoxia (F_i_O_2_ = 13%) [24]. In contrast, in their very well-designed study, Girard et al. [25] demonstrated that hypoxia (F_i_O_2_ = 12.9%) negatively impacts resistance exercise to failure performance at light (30% of 1RM) loads. The reason for the contrasting results can be elucidated by hypoxia exposure duration. The ergolytic effect of hypoxia was proposed to be dependent on exposure duration [43]. In addition, exposure duration, as a moderating factor, was also found to be significant by a meta-analysis [44] that quantifies the effect of acute hypoxia on exercise performance. They concluded that the negative effect of hypoxia increases in parallel with the prolongation of the exercise protocol [44]. In this regard, we can speculate that studies which found no ergolytic effect on repetition to failure performance may not have had enough hypoxia exposure duration during their exercise protocols (~12–15 min) [23,24]. Therefore, it appears that relatively short durations (~20 min) in hypoxia during squat endurance likely influenced the results in our test protocol. Further, bench press endurance test was performed after squat exercise and was negatively affected by hypoxia, during which exposure time was nearly 40 min. This finding may not only be due to the morphological and neuromuscular differences between the lower (squat) and upper (bench press) extremities [18]; it may be due to the prolonged exposure to total hypoxia due to the bench press test being performed after the squat test. Contrary to our argument, Walden et al. [45] reported that various exposure durations (20 min vs. 30 min) to hypoxia (F_i_O_2_ = 13%) had no influence on the bench press and shoulder press endurance. More research is required in this topic to make confirm our speculation.

Females have a greater resistance to hypoxia [46] and recover faster from fatiguing exercise, in contrast to their male counterparts [26]. Latella et al. [47] found that the mechanism in which corticospinal excitability is modulated appears to be sex-specific during resistance training. Lastly, apparent differences between sexes in terms of hormonal status and lean body mass might alter the results for the resistance exercise performance during hypoxia. For this reason, it is difficult to generalize the recommendations from hypoxia studies, all of which were conducted on men [4,18,19,20,23,24,25,38,45]. To the best of our knowledge, this is the first study investigating the effects of hypoxia during resistance exercise in females. In contrast to the aforementioned hypothesis, our results showed that men and women respond similarly to the different doses of hypoxia. In support, previously from our laboratory, sprint interval training performance was not different between sexes both at moderate (2500 m) and high (3500 m) hypoxia [48]. However, in another study, despite similar oxygen desaturation levels, men exhibited higher sympathetic responses to very high hypoxia (F_i_O_2_ = 9.6%) compared to women [49]. Direct comparisons in this regard are not possible, as there is no study directly comparing sexes in a hypoxic resistance exercise setting. However, RPE values in our study were approximately 17–18 at the end of test protocol. We can speculate that near maximal RPE values, in turn, high intensity nature of our test protocol may create a “ceiling effect” that makes any appreciable differences between sexes extremely hard to distinguish, especially during hypoxia.

### 4.3. Cognitive Performance

In our study, cognitive performance was found to be lower in high and moderate hypoxia than low hypoxia and normoxia. These findings are in line with some [50,51], but not all, studies [52,53]. A recent meta-analysis [54] concluded that various characteristics (e.g., cognitive task type, exercise type/intensity, and hypoxia level) moderated the effects of hypoxia and exercise on cognitive function. Similar to our results, acute exposure to severe hypoxia decreased cognitive function compared to moderate and low hypoxic conditions in a dose-dependent manner [55]. Additionally, cognitive function has various domains, such as information processing, executive function, and memory [50,51,52,53,54,55]. The Flanker task which was used in the current study measures executive function that enable an individual to focus attention [36]. Since attentional focus is already reported to moderate acute muscle performance [32,33], decrements in bench press endurance performance during high and moderate hypoxia may be associated with equivalent impairment in cognitive function.

Women have greater resistance to hypoxia and higher SpO_2_ levels than men [30]. In addition, women have greater basal cerebral blood flow than men during hypoxia, possibly associated with estrogen [56]. However, in the current study, we did not observe any sex differences in cognitive performance. This result is consistent with Lefferts et al. [50] suggesting that cognitive performance of the Flanker task was lower in hypoxia (F_i_O_2_ = 12.5%) during 55% HRmax aerobic exercise in both men and women. In contrast, short term (~20 min) severe hypoxia (F_i_O_2_ = 12%) improved the cognitive performance of the Go/NoGo task during exercise, with 45% peak power output in women [57]. Discordance between the previous studies [52,53,57] and ours are related to exercise type (aerobic vs. resistance) and measurement time of cognitive performance (during vs. after) [54].

### 4.4. Heart Rate, RPE, Lactate

Our results demonstrated that high and moderate hypoxia significantly increased RPE and capillary lactate after bench press exercise, but not after squat exercise. Further, heart rate values were not different between conditions. This mechanism can be explained by the following: just as the repetition number of bench press decreases as exposure duration to hypoxia increases, blood lactate responses may also increase due to prolonged exposure to hypoxia.

Our RPE and lactate outcomes are in support of others [4,18,24,45]. In support, RPE and blood lactate were higher in high hypoxia (F_i_O_2_ = 13%) than in moderate hypoxia (F_i_O_2_ = 16%) and normoxia [19]. However, Scott et al. [20] demonstrated no differences in RPE during 5 × 5 repetitions at 80% 1RM during high (F_i_O_2_ = 13%) hypoxia. These conflicting results can be explained by the type of test protocol (circuit vs. traditional vs. repetition to failure). Nevertheless, RPE is a useful tool to determine the intensity of hypoxic resistance exercise, as demonstrated by the current study.

### 4.5. Strengths and Limitations

In the current study, we used a double-blind design. The scientific gold standard design of a double-blind, randomized, crossover trials are rarely carried out in hypoxic resistance exercise studies [1,9,25]. Moreover, we used repetitions to failure, which ensures maximal or near maximal effort and is a potent strategy to induce muscle hypertrophy, possibly due to an increase in metabolic stress [22]. Lastly, for the first time in the literature, we directly compared men and women participants’ acute responses to various doses of hypoxia. However, our study is not without limitations. One of the limitations is that squat and bench exercises are not performed in a random order. Secondly, we did not assess women participants’ menstrual cycle, which may impact physical and cognitive performance [57]. However, the four experimental conditions were completed in nine days, which would have minimized the impact of the menstrual cycle.

## 5. Conclusions

Overall, the current study found that acute hypoxia does not alter resistance exercise responses between sexes differently. More specifically, high (4000 m) and moderate (3000 m) hypoxia significantly decreased bench press endurance and cognitive performance, and significantly increased RPE and blood lactate, as compared to low hypoxia (2000 m) and normoxia. Athletes and coaches should carefully design resistance exercise trainings under high and moderate hypoxia (≥30 min in duration) due to the impaired effects on upper body muscular endurance and perceived exertion. Nevertheless, metabolic stress induced by hypoxic resistance training can improve long term hypertrophic adaptations [40].

## Figures and Tables

**Figure 1 biology-11-00309-f001:**
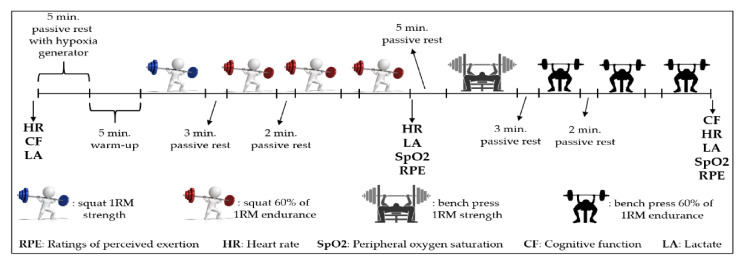
Schematic representation of the experimental sessions.

**Figure 2 biology-11-00309-f002:**
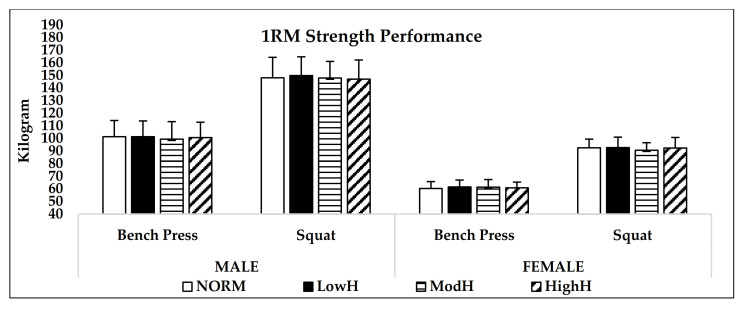
Mean (SD) bench press and squat strength (1-RM) performance. NORM: Normoxia (900 m); LowH: Low dose of hypoxia (2000 m); ModH: moderate dose of hypoxia (3000 m); HighH: high dose of hypoxia (4000 m).

**Figure 3 biology-11-00309-f003:**
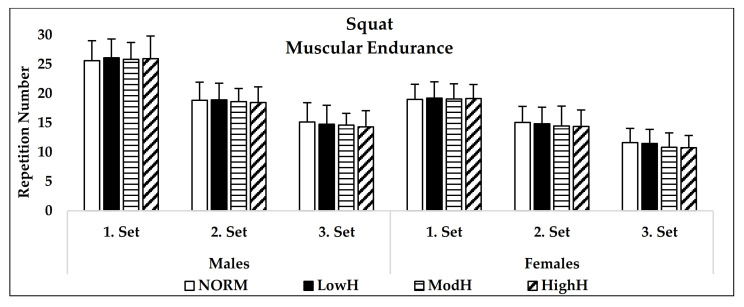
Mean (SD) squat 60% of 1RM endurance performance of males and females. NORM: Normoxia (900 m); LowH: Low dose of hypoxia (2000 m); ModH: moderate dose of hypoxia (3000 m); HighH: high dose of hypoxia (4000 m).

**Figure 4 biology-11-00309-f004:**
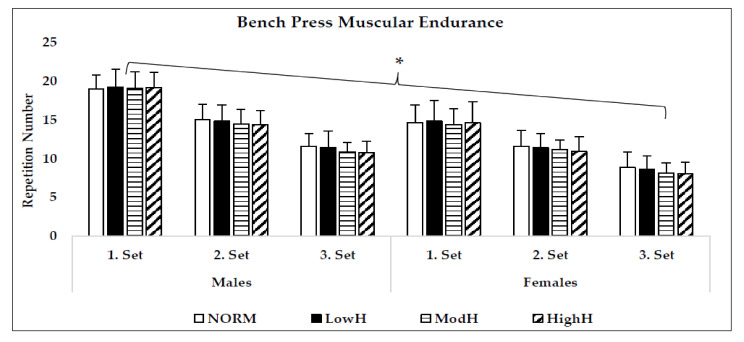
Mean (SD) bench press 60% of 1RM endurance performance of males and females. NORM: Normoxia (900 m); LowH: Low dose of hypoxia (2000 m); ModH: moderate dose of hypoxia (3000 m); HighH: high dose of hypoxia (4000 m). * ModH and HighH were significantly different than NORM and LowH.

**Table 1 biology-11-00309-t001:** Cognitive function parameters.

	Males				Females			
	Pre Test		Post Test		Pre Test		Post Test	
	M	SD	M	SD	M	SD	M	SD
Response Accuracy [%]-Congruent Task								
NORM	96.41	1.4	95.83	1.6	96.58	1.7	96.66	1.3
LowH	96.16	2.3	95.83	2.0	96.41	2.1	97.08	1.3
ModH	96.83	2.3	96.50	1.7	97.16	2.0	96.25	1.9
HighH	96.00	1.8	96.33	1.6	97.33	1.2	95.16	1.6
Response Accuracy [%]-Incongruent Task								
NORM	93.25	2.8	93.33	2.0	94.58	2.4	95.50	1.7
LowH	92.91	1.9	93.16	2.2	95.08	1.3	94.83	1.4
ModH	93.00	3.1	92.33	2.6	94.50	1.7	94.91	1.2
HighH	93.41	1.8	91.16	2.0	94.50	1.5	94.83	1.8
Reaction Time [ms]-Congruent Task								
NORM	493.45	42.2	480.77	46.2	533.40	44.3	517.70	47.7
LowH	496.80	38.9	485.53	46.9	541.04	39.1	518.55	30.3
ModH	497.90	42.8	494.21 *	42.6	530.83	42.4	534.35 *	34.8
HighH	491.95	40.0	502.68 *	35.9	531.91	32.7	542.35 *	38.2
Reaction Time [ms]-Incongruent Task								
NORM	531.81	31.5	516.94	25.3	608.10	49.9	596.34	55.1
LowH	546.70	31.4	528.46 #	34.2	615.10	44.5	588.52 #	45.0
ModH	541.22	27.3	535.80 *	36.6	603.97	51.4	625.48 *	59.1
HighH	542.53	29.7	567.42 *	25.0	607.01	40.3	630.07 *	34.2

NORM: Normoxia (900 m); LowH: Low dose of hypoxia (2000 m); ModH: Moderate dose of hypoxia (3000 m); HighH: High dose of hypoxia (4000 m). *: significantly different than NORM; #: significantly different than HighH.

**Table 2 biology-11-00309-t002:** Lactate, heart rate, ratings of perceived exertion, and oxygen saturation values.

		NORM		LowH		ModH		HighH	
		Lactate (mmol/L)							
		M	SD	M	SD	M	SD	M	SD
Males	Pretest	1.2	0.1	1.23	0.2	1.14	0.1	1.15	0.1
	Postsquat	6.56	1.1	6.45	1.0	6.54	1.1	6.61	1.0
	Postbench	7.11	1.2	7.23 #	1.0	7.98 *	1.2	7.94 *	0.9
Females	Pretest	1.26	0.1	1.2	0.1	1.11	0.1	1.13	0.1
	Postsquat	5.66	0.7	5.7	0.4	5.83	0.4	5.91	0.7
	Postbench	6.01	0.7	6.13 #	0.8	6.58 *	0.5	6.91 *	0.8
		**Heart Rate (beat/min)**							
Males	Pretest	64.75	4.9	63.50	5.31	64.33	3.6	64.58	4.8
	Postsquat	173.25	8.6	171.33	5.39	172.33	4.7	171.33	7.4
	Postbench	176.08	8.4	175.08	10.4	175.33	8.1	176.08	7.6
Females	Pretest	64.75	3.5	60.50	3.3	62.66	4.0	64.16	3.2
	Postsquat	159.83	8.7	157.75	8.4	158.41	9.0	157.91	6.4
	Postbench	162.33	7.8	163.50	8.8	160.83	9.7	160.58	9.2
		**Ratings of Perceived Exertion (6–20)**							
Males	Postsquat	13.41	1.2	13.66 #	1.0	13.58 *	1.1	14.00 *	1.2
	Postbench	15.91	1.9	16.33 #	1.6	17.00 *	2.3	17.66 *	2.1
Females	Postsquat	13.50	1.7	13.83 #	1.5	14.08 *	1.6	13.91 *	1.6
	Postbench	17.41	1.9	17.08 #	2.1	17.75 *	1.9	18.33 *	1.7
		**Oxygen Saturation (%)**							
Males	Postsquat	94.83	1.9	90.75	3.2	85.16	3.7	80.91	2.3
	Postbench	94.33	1.8	89.58	2.9	83.91	3.8	79.83	2.5
Females	Postsquat	95.08	1.5	90.16	2.7	86.41	2.6	81.83	2.4
	Postbench	94.16	2.2	90.08	2.6	85.75	3.9	80.66	2.2

NORM: Normoxia (900 m); LowH: Low dose of hypoxia (2000 m); ModH: Moderate dose of hypoxia (3000 m); HighH: High dose of hypoxia (4000 m). Pretest: before test protocol; Postsquat: Immediately after three sets of squat endurance; Postbench: Immediately after three sets of bench press endurance *: significantly different than NORM; #: significantly different than HH.

## Data Availability

The data presented in this study are available on request from the corresponding author. The data are not publicly available due to restrictions of privacy.

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
