# Peer review of "The Acute Effects of Normobaric Hypoxia on Strength, Muscular Endurance and Cognitive Function: Influence of Dose and Sex"

_biology, 2022, doi:10.3390/biology11020309_

Round 1
Reviewer 1 Report
The aim of the study was to investigate the acute effect of different doses of normobaric hypoxia about maximal, endurance muscular and cognitive performance. I would like to congratulate to the authors. The study is original and interesting.
Abstract
Please, delete abbreviations in the abstract.
Introduction
In spite of the introduction is correct, sometimes the authors speak about long-term changes and another way acute effects. If the hypoxic stimulus is an useful tool, an negative acute effect with for example high dose could impair to future adaptations? Please clarify.
The use of “sex” could be more appropriate in this context. Gender seems a social term. Please review along the text.
Methods
L116 Please, delete “free of any musculoskeletal disorders…” it’s repeated.
L119 Why 900m?
L135-137 Please, include the FiO2 in % and include corresponding to x meters)
Results
L217 What about relative load? I mean, in spite of the total kg in RM is different, maybe the relative kg could elicit new results.
L223 Please, separate the results on endurance muscular.
I suggest to include the data on internal load at the beginning like in the methods are explained previously.
Please, include firstly the significant differences and then non-significant difference in the results. I suggest to summarize because there is too much data.
Discussion
Please, include the aim at the beginning.
L340 SP02… please review.
L346 Please, include in the meters: 4,000 m… Review along the text.
L348 Please, don’t repeated the results.
L411 Some references could be included: DOI: 10.23736/S0022-4707.18.08940-5
The discuss should be clarify. It results difficult to read. Besides, the conclusion is not clear. Why this study could be interesting for coaches?
Author Response
We consider all of your suggestion which can be find in attach, best regards.

Reviewer 2 Report
This paper addressing the effects of exercising in a hypoxic environment on strength and endurance is a topic area that is interesting and I believe that publication of such an area is warranted, however, I have a few concerns regarding the current paper that should be addressed before it could be accepted.
Well the English is generally understandable, there are many places in which the sentence structure or the grammar let the authors down. This needs a significant overhaul in order for the paper to be easily readable.
The literature in the introduction is too focussed on the authors interests and doesn’t take into account the plethora of literature on exercise in hypoxia, this needs to be updated and the authors should avoid cherry picking articles that suit their argument.
For a review on fibre type adaptation following hypoxic exposure please see:
Skeletal Muscle Fiber Type in Hypoxia: Adaptation to High-Altitude Exposure and Under Conditions of Pathological Hypoxia
Change gender to sex
The standard nomenclature for hypobaric hypoxia is HH, so it’s a little confusing reading HH as an abbreviation here and have it mean something else
What altitude did your study take place at and how you calculate the required FiO2 for the gas?
Why was 900m chosen as the normoxic altitude? Was this the university altitude? Can you report the barometric pressure for the testing days please as this is important for the calculation of the FiO2.
Methods.
Please clarify the total time the participants were breathing the hypoxic gas mixture, as this is unclear from Figure 1 and the text.
The results section is confusing and needs to be edited to provide the reader with a clear concise idea of how the outcome measures changed or did not change. The 1rm results is a solid paragraph followed by a series of figures and tables which make several pages completely incomprehensible. This is not a major issue, but the layout needs to be updated so that it doesn’t appear that the authors are relying on the reader to interpret the graphs and guess the results.
Would it be beneficial to the reader if instead of having on a male figure (fig3) that you separated the squatting and bench exercise so that Fig 3 became squat for male and female and thus fig4 became bench for male and female?
Para 3.2 and 3.3 need to be broken up a little more and edited for clarity, currently its very difficult to interpret the results as they are.
Firstly, the discussion needs to have sub headings.
The novelty of the experiment is not clearly highlighted
Line 342, impressive finding is not a scientific term, please reword
Line 348-350 makes no sense
Mixing in the time of exercise duration or hypoxic exposure following this sentence also makes no sense and is not described well as a justification for the results seen prior in the same paragraph.
Line 364. What is PCr? It has never been mentioned before
The discussion around paragraph 360 to 372 is confusion as this was not introduced in the methods or results.
Line 381 no dose effect…this sections argument is unclear to me. It is in contention with the main results and does not build on the mechanistic understanding of the current ms results.
The rest of the paragraph is meandering and needs to be edited.
Line389 on…this section appears to be trying to address the intensity the exercise was conducted at, however it is also not clear, the authors need to formulate the story and then write it clearly, each paragraph is too difficult to read
The authors should also highlight the difference in relative intensity when moving from normoxia to hypoxia, the absolute workload will remain the same, but the relative intensity will increase. This should be discussed
Line 408, why start this sentence with however?
Line 409, why are you providing an explanation for potential differences, while stating that no differences existed between line 406 and 408 at 70% and 30%?
The discussion needs major revision, the structure and thought process leading the reader through the results and inferences is vague and confusing. The authors also discuss several papers that have investigated resistive exercise in hypoxia, but state in the intro that no such studies exists.
I believe this observational study needs to be more cautious with their inferences and conclusions. This discussion is limited and needs to be streamlined in order to highlight the strength of the experimental design.
The limitation section on hypoxic exposure time needs to be explored and the justification of such a statement backed by the literature. This reviewer also does not have a good grasp of the time scale of hypoxic exposure from the methods or results and struggles to link the limitation to the actual study.
Cognitive function is in the title and as far as I can see, only 1 paragraph in the discussion deals with this. This would change, if cognitive function is to remain, then it should also be linked to how it affects the other measured variables.
Conclusion
What do you mean by the intensity of hypoxia?
“The main finding of this study is that the dose of hypoxia affects performance” is not novel and if this is truly the case then the paper unfortunately does not warrant publication.
What clues does your paper provide to coaches? please be more specific
Author Response
We consider all of your suggestions which can be found in attached, thank you very much again for spending time on our manuscript and getting it higher quality. Best regards.

Round 2
Reviewer 1 Report
I would like to congratulate the authors for the improved work. Well done!